# NORAD Tracking of the 2022 February Starlink Satellites and the Immediate Loss of 32 Satellites

Fernando L. Guarnieri[1], Bruce T. Tsurutani[2], Rajkumar Hajra[3], Ezequiel Echer[1], Gurbax S. Lakhina[4]

[1] Instituto Nacional de Pesquisas Espaciais, São José dos Campos, São Paulo, Brazil. (ORCID 0000-0002-8351-6779)
[2] Retired, Pasadena, California USA. (ORCID 0000-0001-7299-9835)
[3] CAS Key Laboratory of Geospace Environment, School of Earth and Space Sciences, University of Science and Technology of China, Hefei, China. (ORCID 0000-0003-0447-1531)
[4] Retired, Vashi, Navi Mumbai 400703, India. (ORCID 0000-0002-8956-486X)

*Correspondence to*: Fernando L. Guarnieri (fernando.guarnieri@gmail.com)

**Abstract.** The North American Aerospace Defense Command (NORAD) tracking of the SpaceX Starlink satellite launch on 2022 February 3 is reviewed. Of the 49 Starlink satellites released into orbit, 38 were eventually lost. Thirty-two of the satellites were never tracked by NORAD. Two different physical mechanisms have been proposed published in Space Weather and one in arXiv to explain the satellite losses. It is argued that none of these three papers can explain the immediate loss of 32 of the 49 satellites. We suggest NORAD satellite tracking information for scientists to further investigate possible loss mechanisms.

## 1 Introduction

Geomagnetic storms (von Humboldt 1808; Gonzalez et al. 1994) are caused by magnetic reconnection (Dungey 1961; Tsurutani and Meng, 1972; Paschmann et al., 1979) between southward interplanetary magnetic fields (IMFs) and the Earth's dayside magnetic fields. The reconnected magnetic fields and solar wind plasma are convected to the midnight sector of the Earth's magnetosphere (magnetotail) where the magnetic fields are reconnected again (Dungey, 1961). The reconnected fields and plasma are jetted from the magnetotail towards the inner magnetosphere (DeForest and McIlwain, 1971), causing auroras (Akasofu 1964) in the midnight sector at geomagnetic latitudes of 65° to 70° and slightly lower (the auroras occur both in the northern and southern polar regions). The auroras also spread to all longitudes covering the Earth's magnetosphere at the above latitudes if the storm is intense and long lasting.

The auroras are caused by the influx of energetic ~10 to 100 keV electrons into the outer regions of the magnetosphere (Anderson 1958; Hosokawa et al. 2020) plus precipitation into the ionosphere causing the diffuse auroras and parallel electric fields above the ionosphere accelerating electrons to ~1 to 10 keV causing the discrete auroras (Carlson et al., 1998). The electrons impact atmospheric atoms and molecules at a height of ~110 to 90 km, excite them and decay giving off auroral lines of violet, green and red light. The influx of the energetic electrons also causes the upwelling of oxygen ions to heights where

they will affect the orbiting satellites, causing enhanced drag on the satellites and eventual lowering of their orbits. This is the standard picture of low altitude satellite drag during magnetic storms.

Three different scenarios were proposed in 2022 to explain the unusual losses of the many Starlink satellites. Before it was known that the Starlink satellites never reached their intended ~500 km altitude, Tsurutani et al. (2022) proposed that prompt penetrating electric fields (PPEFs; Tsurutani et al. 2004, 2007; Lakhina & Tsurutani 2017) could be responsible for those losses. Their Fig. 2 (reshown here as Fig. 2) demonstrated that dayside near-equatorial density increases occurred at ~500 km altitude during the two magnetic storms. However, the present orbital analyses indicate that none of the satellites lost on the first two days reached altitudes higher than 200 km for the entire orbit (they were still in elliptic trajectories). Thus, this loss mechanism must be discarded as a possible cause of the Starlink satellite losses.

The Dang et al. (2022) scenario does not explain such losses in so low latitudes. They used a global upper atmospheric model (TIEGCM) to estimate the Joule heating by Ohmic dissipation at ionospheric altitudes. However, the Joule heating proposed by the authors was more remarkable in high latitudes, while the increases observed in latitudes below 53° were too small to create such an effect. Dang et al. (2022) also predicted losses in 5 to 7 days assuming a constant 210 km satellite altitude. This cannot explain the possible immediate losses of the 32 satellites.

Fang et al. (2022) have used numerical simulations to show 50-125% neutral density enhancements between 200 and 400 km. Their argument is based on effects of Joule heating produced in high latitudes propagating to lower latitudes by large-scale gravity waves with phase speeds of 500 to 800 m s$^{-1}$ (Fuller-Rowell et al. 2008). This propagation may take from 3 to 4 hours and is in addition to the effects of increased UV and EUV fluxes due to the flares. Previous events had taken up to 30 hours for the atmosphere to return to quiet conditions. We note, however, that Fig. 2 in the present work showed that there were very low Joule heating effects in the auroral zone during both of these magnetic storms, thus negating the high latitude Joule heating effects assumed in the model. Pitout et al. (2022) cast doubts that two smallish magnetic storms could have caused the Starlink satellite losses, in agreement with this paper. In 2023, Kakoti et al. (2023) have proposed a new mechanism involving the "combined effects of neutral dynamics and electrodynamic forcing on the dayside ionosphere". They conclude that the minor storms can produce significant ionospheric variations over the American sector, but did not comment on whether these changes could have caused the Starlink satellite losses. None of the above works used the information of the individual Starlink satellite orbits. We have obtained NORAD trackings of many of the individual satellites and will present our findings here. These results should be useful for modelers to understand in more detail the satellite loss mechanisms or for scientists to propose new loss mechanisms that have not been considered before.

## 2 Starlink Launch

On 2022 February 3, at 18:13 UT, SpaceX launched the rocket Falcon 9 Block 5 with the objective of deploying the satellites for the Starlink Group 4-7, the sixth launch to the Starlink Shell 4 (Clark, 2022a, Clark, 2022b). This launch received the international COSPAR identification ID: 2022-010. A video by Manley (2021) illustrates how two stacks of Starlink satellites

could be put into orbit from a single launch vehicle. In this example, each stack of ~30 satellites can be released in different directions. When the satellites separate from this stack, they start individual movements, sometimes colliding gently with others before entering into their individual flight orbits (Manley, 2021). For the February 3 launch, there were two stacks with 24 and 25 satellites in each stack. After the launch, the satellites may be put into edge-on directions with the solar panels parallel to the satellite bodies in an attempt to reduce drag. However, a telecommand is necessary to make them keep the safehold strategy, demanding some time and requiring some minimum antenna pointing (Clark, 2022b; McDowell, 2023).

The SpaceX mission under this study was composed of 49 Starlink satellites that were initially planned to orbit the Earth at ~540 km circular low-Earth orbit (LEO). The initial planned elliptical orbit was 338 km×210 km, at an inclination of 53.22° (Clark, 2022a). Once the initial elliptical orbits were obtained, SpaceX would use onboard propulsion to raise the orbits.

The February 3 deployment of the satellites occurred 15 minutes and 31 seconds after the liftoff, at a release time of 18:28 UT (Clark 2022a, Clark, 2022b). SpaceX considered the launch successful, since the releasing of the satellites occurred in the expected orbits, the rocket stage was recovered as planned, and all the satellites were able to switch to autonomous flight mode.

## 3 Space Weather for the Period

From the time of launch until a day after it, the near-Earth space weather conditions were disturbed with the occurrences of two geomagnetic storms. For the geomagnetic storm identification, we followed the classical definition by Gonzalez et al., (1994), which uses the Dst index (equivalent to the SYM-H index used here) to identify a storm. When the index falls in the range from -30 nT to -50 nT for more than 1 hour, it characterizes a small (weak) storm. Events with decreases between -50 nT and -100 nT for more than 2 hours are considered a moderate storm. An intense storm is one which falls to Dst values of less than -100 nT for more than 3 hours. The recovery phase is the interval when the index begins to return to the values observed prior the storm. It may last from hours to several days. Figure 1 shows the interplanetary and geomagnetic conditions for the period of interest. The solar wind plasma and IMF data at 1 Astronomical Units (AU) are time shifted from the spacecraft location at the L1 libration point ~0.01 AU upstream of the Earth to the nose of the Earth's bow shock. The IMF components are given in the geocentric solar magnetospheric (GSM) coordinate system. The solar/interplanetary data were obtained from the NASA's OMNI database (Papitashvili and King, 2020), and the storm-time SYM-H index from the World Data Center for Geomagnetism, Kyoto, Japan (World Data Center for Geomagnetism et al., 2022).

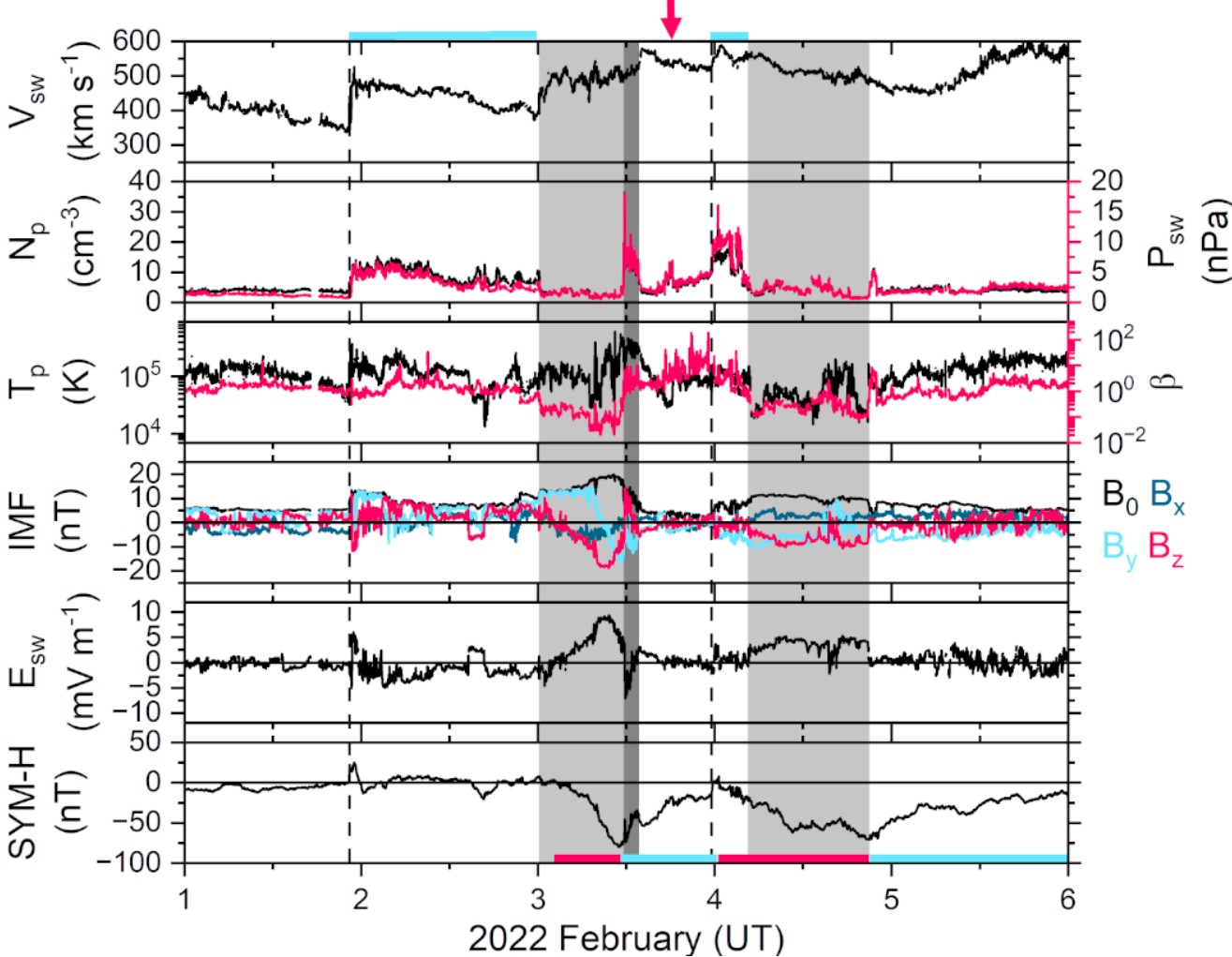

**Figure 1: The interplanetary and geomagnetic conditions during 2022 February 1–5. From top to bottom, the panels are: the solar wind speed $V_{sw}$, the plasma density $N_p$ (black, legend on the left) and ram pressure $P_{sw}$ (magenta, legend on the right), temperature $T_p$ (black, legend on the left), and plasma-$\beta$ (magenta, legend on the right), the interplanetary magnetic field (IMF) magnitude $B_0$ (black), $B_x$ (navy blue), $B_y$ (cian), and $B_z$ (magenta) components, electric field $E_{sw}$ and the geomagnetic SYM-H index. The vertical dashed lines indicate interplanetary fast forward shocks. The light gray shadings indicate magnetic clouds (MCs), and the dark gray**
**shading indicates a solar filament propagated to 1 au. Interplanetary sheaths are marked by green bars at the top. The figure is modified from Tsurutani et al. (2022). The arrow on the top of the figure indicates the Starlink satellites launch time. The red and blue horizontal lines in the Sym-H panel indicate the storm main and recovery phases, respectively.**

A few days prior to the Starlink satellite launch, on January 29, at ~23:00 UT, a M1.1 solar flare erupted from the active region
AR 2936. The flare can be seen in GOES X-ray data as a sudden increase in the radiation flux (NOAA, 2022). The activity in

AR 2936 led to the release of a coronal mass ejection (CME) at 23:36 UT. Using the observed CME velocity near the sun, the

arrival of the CME at 1 AU was predicted. The geomagnetic impact of the interplanetary counter part of the CME or the

interplanetary CME (ICME) was the occurrence of a moderate intensity magnetic storm (Gonzalez et al. 1994; Echer et al.

2008) with a peak SYM-H intensity of –80 nT on February 3. A second moderate intensity geomagnetic storm with a SYM-H intensity of –71 nT occurred on February 4.

The speed of the ICME at 1 AU was ~500 km s$^{-1}$. This is classified as a moderately fast ICME (faster than the slow solar wind speed of ~350 to 400 km s$^{-1}$). The fast ICME acted as a piston and caused an upstream shock and a sheath. The upstream fast forward shock reached the Earth at ~22:19 UT on February 1 (indicated by a vertical dashed line). The shock/sheath caused a sudden impulse (SI+) of 22 nT at the Earth's surface (Joselyn and Tsurutani, 1990; Araki, 1994; Tsurutani and Lakhina, 2014, Fiori et al., 2014; Lühr et al., 2009; Oliveira & Samsonov, 2018; Takeuchi et al., 2002) which is noted as an increase in the SYM-H index the high sheath ram pressure (in front of the fast ICME) compressed the Earth's magnetosphere). The sheath following the shock did not contain major southward IMFs, so was generally not geoeffective (other than creating the SI+ and magnetospheric compression). The magnetic cloud (MC) portion of the ICME is identified (Burlaga et al. 1981; Burlaga et al. 1998; Tsurutani et al. 1988; Gopalswamy et. al 1998; Lepri and Zurbuchen, 2010; Sharma and Srivastava, 2012; Gopalswamy, 2015, 2022) by high IMF magnitude $B_0$ and low plasma-β (the ratio between the plasma thermal pressure and the magnetic pressure), and is shown by a light gray shading. The MC extends from ~23:54 UT on February 2 to ~13:44 UT on February 3. The IMF $B_z$ component of the MC has the characteristic "fluxrope" configuration with a southward component followed by a northward component. A "fluxrope" is the geometry of magnetic fields where field aligned currents are flowing within the "magnetic rope" (Russell and Elphic, 1979; Yashiro et al., 2013; Marubashi et al., 2015).

During the southward IMF interval, the SYM-H index decreased to a peak value of –80 nT at ~10:56 UT on February 3. Thus, the magnetic storm is caused by the magnetic reconnection process between the interplanetary field and the Earth's magnetopause fields (Dungey 1961). The dark gray shaded region is the high-density solar filament portion of the ICME (Illing & Hundhausen 1986; Burlaga et al. 1998; Wang et al., 2018; Kozyra et al., 2013). The filament also causes a compression of the magnetosphere and a sudden increase in the SYM-H index to –39 nT.

A second fast forward shock is identified at ~23:37 UT on February 3 (marked by a vertical dashed line). The shock caused a SI+ of ~17 nT. The following sheath did not contain any major IMF southward component, so again it was not geoeffective. The MC portion of the second ICME is indicated by a light gray shading from ~04:37 UT to ~21:02 UT on February 4. The MC had a peak IMF $B_0$ of ~12 nT at ~08:02 UT. The MC $B_z$ component profile is different from the previous MC. $B_z$ is negative or zero throughout the MC. The negative $B_z$ causes the second magnetic storm of peak intensity -71 nT at ~20:59 UT on February 4. There was no solar filament during this second ICME event. From Fig. 1, it is clear that SpaceX launched their Starlink satellites into a moderate intensity magnetic storm. At the present time it is unclear what the solar source (solar flare or prominence eruptions; Tang et al., 1989) was for this second ICME.

The effects of these storms on the atmospheric mass density are analyzed using data from the Swarm B satellite (Fig. 2). The Swarm mission is operated by the European Space Agency (Swarm, 2004). Swarm B is in a circular orbit at ~500 km, with an inclination of ~88° and orbital period of ~90 minutes, so there are about 15 orbits per day. The orbits have been numbered for each day. At 00:00 UT on February 2, the satellite was at ~–10° latitude at ~09:00 local time (LT) on the dayside, and was moving towards the south pole. The mass density is ~3.5×10$^{-13}$ kg m$^{-3}$ (a light blue color). Continuing in time as the orbit

crosses over the south pole and enters the nightside ionosphere at ~20:00 LT, it is noticed that between –53° and +53° the density reduces to ~1.5×10$^{-13}$ kg m$^{-3}$ (a dark blue color). The other orbits on February 2 show a similar pattern between the nightside and dayside passes.

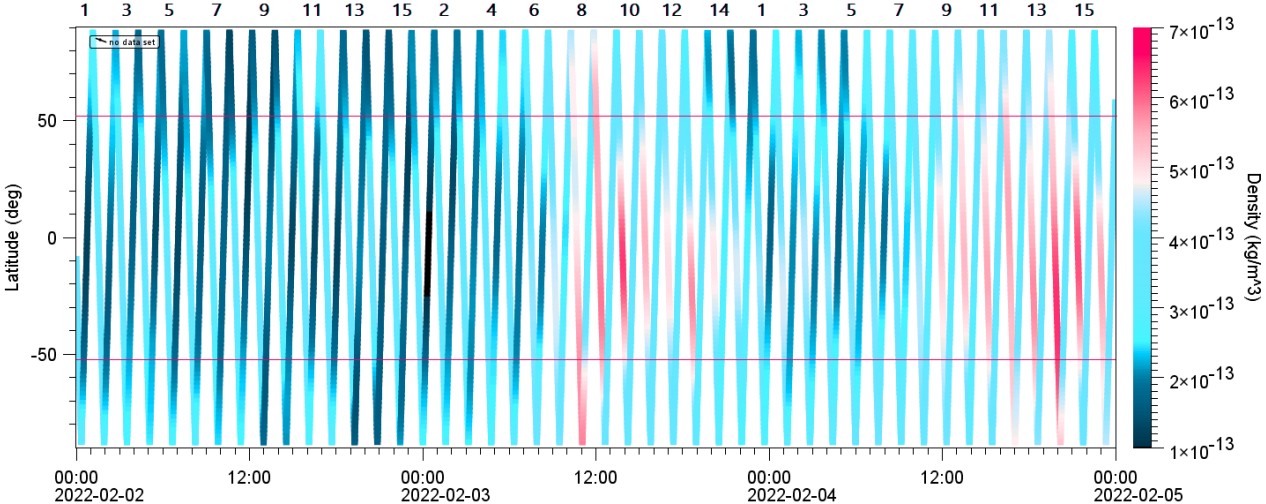

**Figure 2: The Swarm B mass impact data for February 2–4. The mass density is shown as a function of UT (x-axis) and geographic latitude (y-axis). It can be noted that the observations cover both day (north-to-south hemispheric passes) and night (south-to-north hemispheric passes) sides of the globe. February 2 was a quiet day before the two magnetic storms and is shown as a "quiet-day reference". The mass density values are given in linear color scale on the right. Two red horizontal lines at +53° and –53° indicate the upper limits of the intended Starlink satellite orbits. Swarm B orbits on each day from the north pole to the south pole and back are marked by numbers from 1 to 15. Partial orbit 1 for February 2 is shown at the beginning of the figure.**

On orbit 8 of February 3, there is the first sign of a change (increase) in the mass impact at middle and low latitudes (~5.0×10$^{-13}$ kg m$^{-3}$, an orange color). This occurs at the south pole crossing at ~10:00 UT, just before the peak of the first magnetic storm. There is a density enhancement (red coloration) throughout this downward dayside pass, across the magnetic equator and to the south pole. There is a local maximum of density at ~14:00 UT and ~09:00 LT at 10° latitude. On orbits 9–13 of February 3, the predominant density enhancements are on the dayside passes in the equatorial and midlatitude ranges. The enhancements are larger than those at higher latitudes. The maximum density of ~5.5×10$^{-13}$ kg m$^{-3}$ occurred at ~19:00 UT and ~09:00 LT. This represents a density peak increase of ~50% relative to the quiet daytime density (February 2).

On orbits 9–13 of February 3, the nightside equatorial and midlatitude densities are ~3.5×10$^{-13}$ kg m$^{-3}$. This is higher than the February 2 (quiet time) nightside densities of ~1.5×10$^{-13}$ kg m$^{-3}$. Thus, during the magnetic storm, the nightside peak densities increased by ~130%. It is noted that the nighttime peak densities are less than the daytime peak densities. This latter feature will be explained later in this paper.

The high impact mass (red color) fades out by the end of February 3 and does not start again until orbit 8 of February 4. A density peak of ~5.3×10$^{-13}$ kg m$^{-3}$ at the equatorial region on orbit 8 occurred at ~12:00 UT. This is approximately 10 hours

after the slowly developing second magnetic storm started at ~00:15 UT on February 4. From orbit 8 to 11 the predominant density enhancement occurs at the equatorial to middle latitudes with little or no enhanced impact in the auroral/polar regions. The maximum density of ~$6.3\times10^{-13}$ kg m$^{-3}$ occurred at 20:00 UT on dayside pass 14, and extended from ~ $-15°$ to $-60°$ latitudes. The peak time is coincident with the peak in the second magnetic storm. On passes 15 and 16, the density decreases, and the enhanced density occurs mainly at the equator and middle latitudes. The maximum density during this second storm event was ~80% higher than the dayside density values detected on February 2.

The nightside density on orbit 14 on February 4 was ~$4.3\times10^{-13}$ kg m$^{-3}$. This is ~190% higher than the quiet time value on February 2. The nighttime peak densities are lower than the daytime peak densities, similar to the first storm features. The data for February 5 and 6 look similar to the quiet day interval of February 2, so are not shown to conserve space.

## 4 Magnetic Storm Effects on Starlink Satellite Survivability

Among the 49 released satellites, only 17 could be tracked by the North American Defense Command (NORAD) some days later. Thirty-two satellites were never listed by NORAD, thus we assume that they were immediately lost after launch. This may have happened due to problems in tracking them (due to extremely fast orbital decays in the first hours after the release or due to substantially different satellite positions than expected for the launch).

Considering the events since the start of the deployment of the Group 4 satellites, in 2021 November, the Starlink launch efficiency have been around 97.5% successful for the last 75 launches to date. However, for the launch being analyzed here, Starlink Group 4-7, represents a significant reduction in this efficiency, with an orbit insertion failure rate of 77.6%. Figure 3 shows the percentage of loss for the Starlink launches #33 up to #107. The event analyzed here was marked as the red column in the plot. The data used for this plot is available in Table 1 (Appendix) with the statistics of the launches for the 75 most recent Starlink satellite launches, up to 2023 September. For the calculation of failure percentage, only satellites that failed during the orbit injection process were considered.

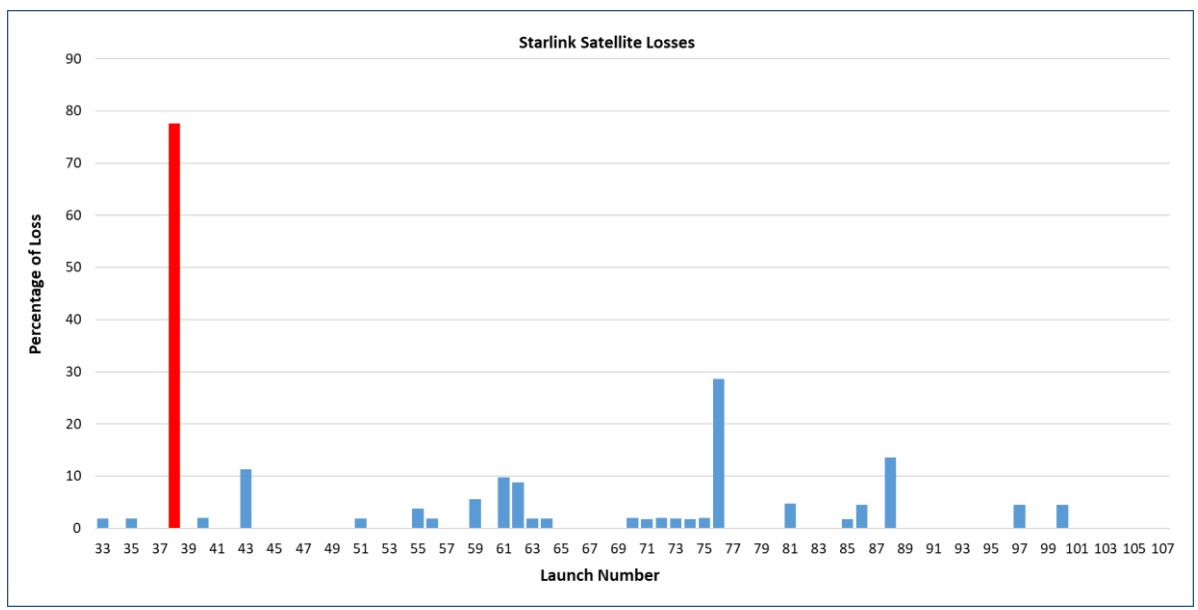

Figure 3: Percentage of Loss for Starlink lauches #33 to #107. The event of February 05, 2022, marked as the red column in this plot, shows a failure rate of 77.6%. The second highest peak, launch #76, shows an unusual high loss event since this launch (Group 6-1) included several changes compared with the previous missions (first launch of larger Starlink V2 Mini satellites, it was the first use of an Argon-fuelled Hall-effect thruster in space, and changes in the tension rods to avoid releasing them in space).

On February 5, a first group of 4 Starlink satellite tracking was made available by NORAD. Two more satellites were tracked on February 07 and 08. All these satellites had very low perigees, ~200 km altitude. The apogees were also very low, always below 350 km, and in some cases as low as 250 km. Since the orbit injection velocities were too low for such unexpected low orbits, these satellites did not survive long and all of them reentered within a few days.

A second group of satellites, formed by 11 satellites, was tracked some days later, on February 8. These satellites were able to
perform their ascending movements, changing from elliptical to circular orbits, and rising to higher and more stable intermediate orbits at ~350 km. The satellites were kept in this position for a few days. Afterwards their orbits were boosted to their final altitudes of ~540 km.  However, one of these satellites, Starlink-3165 (NORAD number 51471), showed communication problems beginning on 2022 October 31. The satellite continued being tracked in flight but out of control until finally deorbited October 17, 2023. The cause of this communication failure is still undisclosed, and thus it is not possible to
verify whether it could be related to the problems experienced during the first hours/days after launch.

**5 Satellite Tracking Timeline**

In order to make it easier to understand all the sequence of events, a timeline was created with the space weather events, individual satellite tracking and other available information. Figure 4 shows this timeline. The satellites are identified by their NORAD numbers. Only those tracked after February 8 were linked to their Starlink numbers.

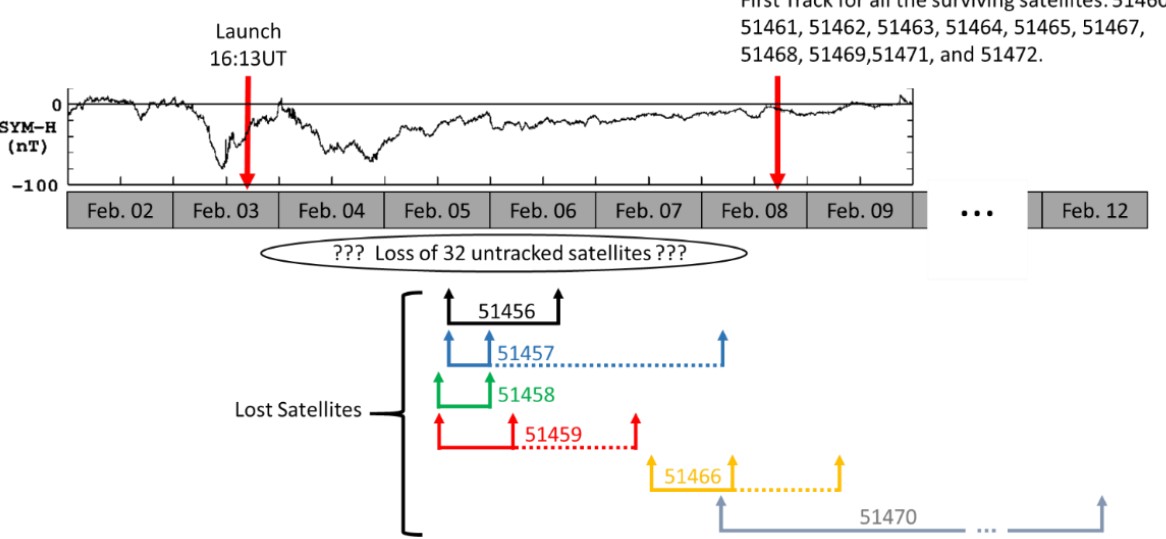

**Figure 4: Timeline for the satellite tracking occurring between February 2 and 12, 2022. The plot shows the SYM-H for the period from February 02-09, 2022. On the top of the plot, a red downward pointing arrow indicate the launch time, and a second arrow indicates the beginning of the tracking of the 11 surviving satellites. At the bottom of the plot, the upward arrows in different colors indicate the beginning and end times of the tracking for each lost satellite. The oval mark indicates the time interval when the 32 lost satellites were expected to be tracked.**

Figure 4 shows a plot of the SYM-H index which indicates the geomagnetic disturbances and the occurrence of geomagnetic storms. The two storm peaks are: SYM-H = –80 nT on February 3 and SYM-H = –71 nT on February 4. The red downward pointing arrows indicate the launch times, and the beginning of the tracking of the 11 surviving satellites, respectively. It should be noted that the Starlink satellites were launched in the recovery phase of the first storm (SYM-H increasing from its minimum value). Thus, the satellites are expected to have experienced effects from the first magnetic storm. It is also noticed that the second storm main phase started at the beginning of February 4 and continued for almost the entire day. Any Starlink satellites surviving the first storm would experience the effects of the second storm as well.

At the bottom of Fig. 4, the upward arrows indicate the beginning and end times of the tracking for all other lost satellites (besides the original 32 satellites never tracked). For satellites 51457, 51459, and 51466, the extended dashed line and another arrow indicate the "official" decay times. An oval mark indicates the time interval when the 32 satellites were expected to be tracked, but were already lost.

## 6 Surviving Satellite Orbits Information

Figures 5 and 6 show the orbit information for the decayed satellites and for the operational satellites, respectively.

In Fig. 5 the vertical axes give the satellite altitudes and the horizontal axes give the tracking sequences. The two dashed black lines indicates the perigees and the apogees expected for the satellite launches, 210 km and 338 km, respectively. The red and

blue lines indicate the apogee and perigee at each tracking point. The date and time of the first and last tracking is indicated under the horizontal axis.

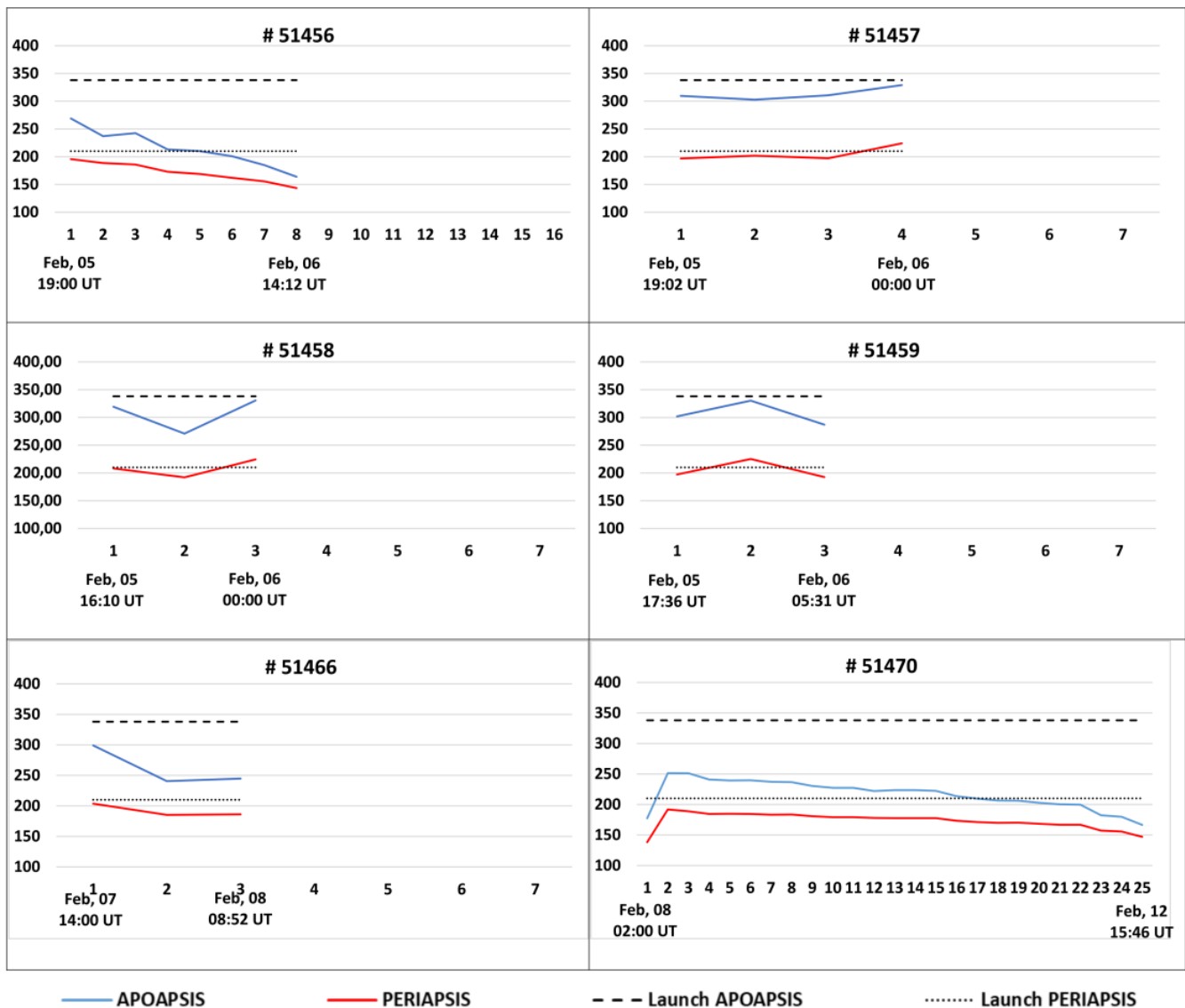


**Figure 5: Panels showing the orbit perigees (red lines) and apogees (blue lines) for each of the decayed satellites. The vertical axis in each panel gives the satellite height, and the horizontal axis indicates the tracking sequence. The dates and times under the horizontal axis indicate the time of the first and the last tracking. The two dashed black lines indicate the perigees and the apogees for the launch.**


For all the above cases, the satellites were in very low orbits in the first track, close to the lowest orbits expected for the lowest perigees. The apogees were always very far (lower) from the expected values for the launch, and sometimes even closer to the values expected for perigees.

It can be noted that some satellites started to rise in altitude, but most likely were lost due to insufficient thrust in such low orbits with increased atmospheric drag.

A contrasting scenario is shown in Fig. 6. All of these satellites survived the launching episode. After initial tracking by NORAD (all of them starting on 2022 February 8) they were boosted by onboard propulsion to safer (higher) altitudes.

The plots are in the same format as in Fig. 5, but the horizontal axes now indicate the initial tracking (on February 8) until March 30.

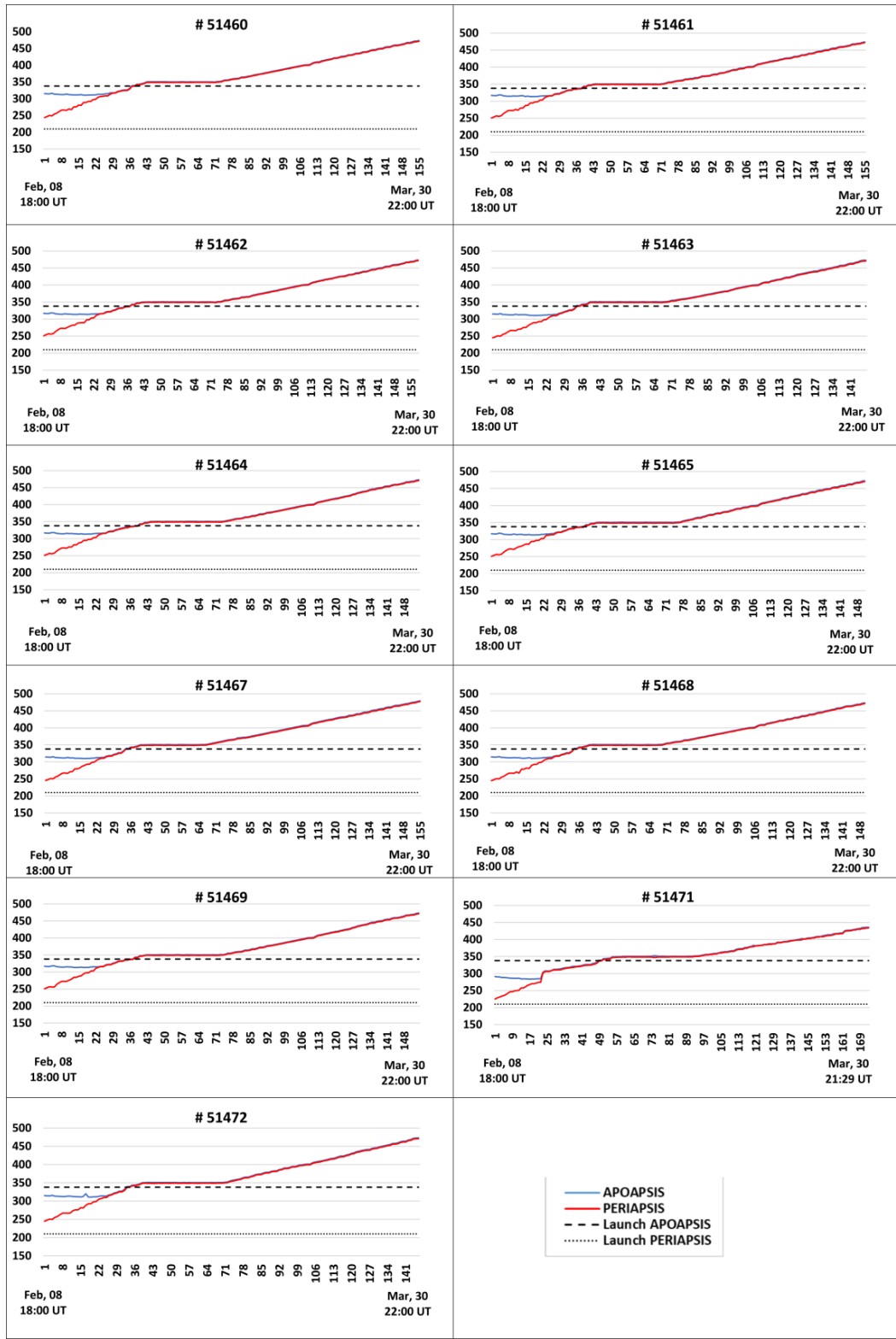

**Figure 6: Panels showing the orbit perigees (red lines) and apogees (blue lines) for the surviving satellites. The vertical axes give the satellite altitudes (in km), and the horizontal axes indicate the tracking sequences from 2022 February 8 to March 30. The two dashed black lines indicate the perigees and apogees expected at the time of the launch.**

It is interesting to note from Fig. 6 that all of the satellites started their orbits in elliptical configurations, with apogee and perigee values much higher than the (decayed) satellites shown in Fig. 5. The Fig. 6 satellite orbits were very close to the specified values for the launch.

The orbit shapes changed to circular configurations (indicated by the merging of the red and blue lines) with subsequent altitude increases to intermediate orbital configurations. The rising to the final orbits were done very slowly, and none of the satellites

had reached the final ~540 km altitude originally envisioned by March 30, almost two months after the launch.

## 7 Discussion and Conclusions

We have shown the available SpaceX Starlink satellite orbital plots as well as the sequence of events observed. The NORAD system was never able to identify 32 satellites. They were presumably lost between a few hours to a few days after launch. This implies possible quite heavy drag in the equatorial to midlatitude (up to 53° latitude) regions of the atmosphere at ~200

km altitude. At the present time there is not a known mechanism to cause such strongly enhanced drag at such low latitudes and altitudes.

Some of the satellites did survive the dual storm event. Since all the Starlink satellites were launched at the same time and at the same altitude, and they had such widely varying fates (some being immediately lost, some surviving) it is clear that each one had a different flight history. This may have to do with the orientation of the satellite during its release (unknown), density

pockets affecting stronger drag, or even satellite-satellite collisions during the release process. Electrostatic Discharges (ESD) were not considered since SpaceX mentioned the satellites were functional and communicating until the reentering in the atmosphere.

It took several more days for NORAD to make available the tracking of another train of 11 other satellites. The latter satellites were in more favorable positions (altitudes), allowing their recovery and rise to more stable orbits.

One can note from the above discussion that different satellites had extremely different orbital decay rates, indicating that one scenario cannot fit all 43 satellite cases. In particular, we are most concerned about the possible losses of 32 of the satellites within the first 48 hours of launch such that they could never be tracked by NORAD.

As mentioned in the introduction, Tsurutani et al. (2022) proposed that prompt penetrating electric fields (PPEFs; Tsurutani et al. 2004, 2007; Lakhina & Tsurutani 2017) could be responsible for those losses. This hypothesis was discarded since the

ionospheric plasma density percent changes by PPEFs are effective in much higher altitudes, ~500 km. We now know that the Starlink satellites never reached such altitudes. However, it was shown for the first time using the Swarm satellite deceleration data that storm time PPEFs may be a main loss mechanism for satellites orbiting at ~400 to 500 km altitudes. A referee of this paper asked the question, could this PPEF mechanism cause high enough densities at lower (~200 km) altitudes to create severe

satellite drag? We think this is not probable in such a low altitude, however, computer modeling will be useful to determine if a similar effect or a change in the local dynamics could occur around 200 km.

Dang et al. (2022) used a global upper atmospheric model (TIEGCM) to estimate the Joule heating by Ohmic dissipation at ionospheric altitudes, but expecting losses in 5 to 7 days assuming a constant 210 km satellite altitude. However, the predicted loss time scales cannot explain the fast decay of the satellites in the first hours or days.

Fang et al. (2022) have used numerical simulations to estimate increases in neutral density between 200 and 400 km in high latitudes due to Joule heating. In the Fang et al. model, the large-scale gravity waves (Fuller-Rowell et al. 2008) would propagate the effects to lower latitudes. However, there were very low Joule heating effects in the auroral zone during both of these magnetic storms, thus negating the high latitude Joule heating effects assumed in the model.

Walach and Grocott (2019) using SuperDARN radar data showed that during geomagnetic storms, ionospheric convection may expand to latitudes as low as 40° magnetic latitude, in the ionospheric F-region (200 – 300 km altitude). This latitude is lower than previously assumed as the limit for the convection from the polar cap (typically 50° magnetic latitude). Although it is possible the satellites may have crossed these regions when travelling through their highest latitude in the orbits, this would be relatively short intervals of time compared with their orbital periods. We expect this effect will be minor.

Kakoti et al. (2023) have suggested that "significant morning-noon electron density reductions elucidated storm-induced equatorward thermospheric wind which caused the strong morning counter electrojet by generating the disturbance dynamo electric field. Substorm related magnetospheric convection resulted in significant noon-time peak in equatorial electroject on 4 February". This is a very interesting possible explanation. But can it explain the near-immediate loss of 32 of the Starlink satellites?

Most of the already published and under review articles about these events were based in modeling and simulations. This event represents a unique opportunity to have in situ observations from 49 satellites that experienced different conditions and fates. Although is well known that private companies restrict the sharing of their telemetry data in order to preserve their technology, if Starlink could make public a minimum dataset of telemetry data, it would allow a multi-point data series analysis useful to understand the physics behind this event. A dataset could give the position for each satellite, their velocities and altitudes, and some indication of whether the propulsion was active, and the solar panel's position. Even knowledge of a few satellites would be extremely useful.

Knowing the position of the satellites and their velocity profiles would indicate the region (in local time) where the most intense drag increases occurred. That could be in the midnight sector, due to some effect of the magnetosphere tail reconnection, although not expected in such low latitudes for a moderate storm. The increases in the dayside could indicate some change in the ionosphere induced by electric field penetration (affecting not only the ionosphere, but with possible effects on neutrals as well). Tsurutani et al. (2004) had already demonstrated the PPEF occurrence in higher altitudes (around 500 km). However, the plasma drag to higher altitudes may lead to the repopulation of other ions or even neutrals in lower altitudes. Even the possibilities of collisions between satellites during the first moments after launch could be confirmed or discarded by a quick analysis of their accelerometer/velocity data, and whether the release direction played a role in the satellite fate.

The velocity profiles along these inclined orbits could show us, by the comparison among the satellites, whether there is a propagation of effects from high latitudes in the polar region to lower/mid latitudes. This analysis could also confirm the observations by Walach and Grocott (2019) that during the geomagnetic storms auroral fluxes may expand equatorward to as low as 40°, lower than previously expected as limited to 50° and higher. Also, these observations would help to understand whether these regions with increased drag were continuous regions or are "tongues" of increased densities, and whether these phenomena could spread in all local times or are restricted to specific regions.

The present work makes it clear that the standard satellite drag computer codes are not able to predict the fast loss of 32 of the Starlink satellites, perhaps including the possible nonlinearity of the atmospheric changes during disturbed geomagnetic periods. Information of the satellite drag with spacecraft orientation would help improve current models. Present models assume satellites with standard cross sections. However, the Starlink satellites are formed by two flat panels (the satellite body and the solar panel). Just after the launch, the solar panel is folded on top of the satellite panel. During the period for the orbital elevation, the solar panel is unfolded but kept aligned with the satellite body, becoming a long flat panel. In the final position, the satellite assumes an "L" shape. This unusual shape of Starlink satellites makes their cross sections completely different from traditional satellites.

Future computer codes should take these different shapes and orientations into account, and then calculate a maximum and minimum time for reentry. The codes should also consider satellite tumbling. Would the loss times be within the maximum and minimum entry times or would they be even greater?

## 8 Final Comments

The losses of the February 2022 Starlink satellites were quite varied. Different satellites were lost at different times. Some satellites even survived. Clearly a simple statement of a value of enhanced drag is insufficient to explain the enormous variability in the different satellite responses. The most difficult problem is explaining the loss of 32 satellites within the first 2 days after launch. At this time, we do not have a physical explanation that involves the two magnetic storms. We hope to stimulate the scientific community to search for currently unknown physical mechanisms that might be able to explain such enormous drag occurring near equatorial regions at low altitudes. Such an event is a rare opportunity to have multi-point local data, during a disturbed period, and in a region where the satellites usually stay for a very short period of time before being propelled to their final and higher orbits.

It is also a remote possibility that the immediate 32 satellite losses were due to satellite-satellite collisions instead of or precipitated by increased drag during the magnetic storms.

The main result from this study is that even a moderate storm may lead to losses when dealing with such low altitudes (at the limit of orbital conditions). The scientific community would greatly benefit if companies operating satellites in these low orbits could make public a "scientific dataset" derived from their telemetry data (while protecting their sensitive information). The

large number of satellites, almost evenly distributed across different latitudes and local times, as well as various altitudes (in a region where the satellites typically remain for a short period before being propelled to their final, higher orbits), would allow the mapping of the effects in their spatial distribution and their evolution (drift) over time. This would improve preparedness against space weather events and enhance our understanding of the physics in the ionosphere/mesosphere region during disturbed geomagnetic periods.

## Competing interests

One of the co-authors is a member of the editorial board of Nonlinear Processes in Geophysics.

## Acknowledgments

The work of R.H. is funded by the Chinese Academy of Sciences "Hundred Talents Program". E.E. thanks Brazilian CNPq for research grant (contract PQ-301883/2019-0). Thanks to the Brazilian Space Agency (AEB) and the Ministry of Science,
Technology and Innovation.

## Data Availability Statement

The solar/interplanetary data were obtained from the NASA's OMNI database (Papitashvili and King, 2020), and the storm-time SYM-H index from the World Data Center for Geomagnetism (World Data Center for Geomagnetism et al., 2022. The Swarm mission is operated by the European Space Agency (Swarm, 2022). Swarm data is publicly available at https://swarm-
diss.eo.esa.int. Starlink satellite launches information is accessible at Wikipedia (2022) and McDowell (2023).

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

**APPENDIX**

Table 1. Statistics on Starlink satellite launches from 2021 November (when Group 4 began to be deployed) to 2023 September 12, with the events ordered according to the launch date. The events marked with "*" indicate launches with another satellite in a rideshare configuration. The event marked with "**" indicates the launch Group 6-1, that included several changes compared with the previous missions (it was the first launch of larger, upgraded Starlink V2 Mini satellites with four times the bandwidth of previous models; it was the first use of a Argon-fuelled Hall-effect thruster in space. Space-X also made changes in the tension rods to avoid releasing them in space) that may have increased the risk and number of failures. The Table information was taken from McDowell (2023) and Wikipedia (2022).

| Mission | Launch Number | Launch Date (Year-DOY) | Number of Satellites | Early Deorbit | Failure (%) |
|---|---|---|---|---|---|
| Starlink Group 4-1 | 33 | 2021-104 | 53 | 1 | 1.9 |
| Starlink Group 4-3 | 34* | 2021-115 | 48 | 0 | 0.0 |
| Starlink Group 4-4 | 35 | 2021-125 | 52 | 1 | 1.9 |
| Starlink Group 4-5 | 36 | 2022-001 | 49 | 0 | 0.0 |
| Starlink Group 4-6 | 37 | 2022-005 | 49 | 0 | 0.0 |
| Starlink Group 4-7 | 38 | 2022-010 | 49 | 38 | 77.6 |
| Starlink Group 4-8 | 39 | 2022-016 | 46 | 0 | 0.0 |
| Starlink Group 4-11 | 40 | 2022-017 | 50 | 1 | 2.0 |
| Starlink Group 4-9 | 41 | 2022-022 | 47 | 0 | 0.0 |
| Starlink Group 4-10 | 42 | 2022-025 | 48 | 0 | 0.0 |
| Starlink Group 4-12 | 43 | 2022-029 | 53 | 6 | 11.3 |
| Starlink Group 4-14 | 44 | 2022-041 | 53 | 0 | 0.0 |
| Starlink Group 4-16 | 45 | 2022-045 | 53 | 0 | 0.0 |
| Starlink Group 4-17 | 46 | 2022-049 | 53 | 0 | 0.0 |
| Starlink Group 4-13 | 47 | 2022-051 | 53 | 0 | 0.0 |
| Starlink Group 4-15 | 48 | 2022-052 | 53 | 0 | 0.0 |
| Starlink Group 4-18 | 49 | 2022-053 | 53 | 0 | 0.0 |
| Starlink Group 4-19 | 50 | 2022-062 | 53 | 0 | 0.0 |
| Starlink Group 4-21 | 51 | 2022-076 | 53 | 1 | 1.9 |
| Starlink Group 3-1 | 52 | 2022-077 | 46 | 0 | 0.0 |
| Starlink Group 4-22 | 53 | 2022-083 | 53 | 0 | 0.0 |
| Starlink Group 3-2 | 54 | 2022-084 | 46 | 0 | 0.0 |
| Starlink Group 4-25 | 55 | 2022-086 | 53 | 2 | 3.8 |
| Starlink Group 4-26 | 56 | 2022-097 | 52 | 1 | 1.9 |
| Starlink Group 3-3 | 57 | 2022-099 | 46 | 0 | 0.0 |
| Starlink Group 4-27 | 58 | 2022-101 | 53 | 0 | 0.0 |
| Starlink Group 4-23 | 59 | 2022-104 | 54 | 3 | 5.6 |
| Starlink Group 3-4 | 60 | 2022-105 | 46 | 0 | 0.0 |
| Starlink Group 4-20/SLTC | 61* | 2022-107 | 51 | 5 | 9.8 |
| Starlink Group 4-2/BW3 | 62* | 2022-111 | 34 | 3 | 8.8 |
| Starlink Group 4-34 | 63 | 2022-114 | 54 | 1 | 1.9 |
| Starlink Group 4-35 | 64 | 2022-119 | 52 | 1 | 1.9 |
| Starlink Group 4-29 | 65 | 2022-125 | 52 | 0 | 0.0 |
| Starlink Group 4-36 | 66 | 2022-136 | 54 | 0 | 0.0 |
| Starlink Group 4-31 | 67 | 2022-141 | 53 | 0 | 0.0 |
| Starlink Group 4-37 | 68 | 2022-175 | 54 | 0 | 0.0 |
| Starlink Group 5-1 | 69 | 2022-177 | 54 | 0 | 0.0 |
| Starlink Group 2-4 | 70 | 2023-010 | 51 | 1 | 2.0 |
| Starlink Group 5-2 | 71 | 2023-013 | 56 | 1 | 1.8 |
| Starlink Group 2-6 | 72* | 2023-014 | 49 | 1 | 2.0 |
| Starlink Group 5-3 | 73 | 2023-015 | 53 | 1 | 1.9 |
| Starlink Group 5-4 | 74 | 2023-020 | 55 | 1 | 1.8 |
| Starlink Group 2-5 | 75 | 2023-021 | 51 | 1 | 2.0 |

| | | | | | |
|---|---|---|---|---|---|
| Starlink Group 6-1 | 76** | 2023-026 | 21 | 6 | 28.6 |
| Starlink Group 2-7 | 77 | 2023-028 | 51 | 0 | 0.0 |
| Starlink Group 2-8 | 78 | 2023-037 | 52 | 0 | 0.0 |
| Starlink Group 5-5 | 79 | 2023-042 | 56 | 0 | 0.0 |
| Starlink Group 5-10 | 80 | 2023-046 | 56 | 0 | 0.0 |
| Starlink Group 6-2 | 81 | 2023-056 | 21 | 1 | 4.8 |
| Starlink Group 3-5 | 82 | 2023-058 | 46 | 0 | 0.0 |
| Starlink Group 5-6 | 83 | 2023-061 | 56 | 0 | 0.0 |
| Starlink Group 2-9 | 84 | 2023-064 | 51 | 0 | 0.0 |
| Starlink Group 5-9 | 85 | 2023-065 | 56 | 1 | 1.8 |
| Starlink Group 6-3 | 86 | 2023-067 | 22 | 1 | 4.5 |
| Starlink Group 2-10 | 87 | 2023-078 | 52 | 0 | 0.0 |
| Starlink Group 6-4 | 88 | 2023-079 | 22 | 3 | 13.6 |
| Starlink Group 5-11 | 89 | 2023-083 | 52 | 0 | 0.0 |
| Starlink Group 5-7 | 90 | 2023-088 | 47 | 0 | 0.0 |
| Starlink Group 5-12 | 91 | 2023-090 | 56 | 0 | 0.0 |
| Starlink Group 5-13 | 92 | 2023-094 | 48 | 0 | 0.0 |
| Starlink Group 6-5 | 93 | 2023-096 | 22 | 0 | 0.0 |
| Starlink Group 5-15 | 94 | 2023-099 | 54 | 0 | 0.0 |
| Starlink Group 6-15 | 95 | 2023-102 | 15 | 0 | 0.0 |
| Starlink Group 6-6 | 96 | 2023-105 | 22 | 0 | 0.0 |
| Starlink Group 6-7 | 97 | 2023-107 | 22 | 1 | 4.5 |
| Starlink Group 6-8 | 98 | 2023-113 | 22 | 0 | 0.0 |
| Starlink Group 6-20 | 99 | 2023-115 | 15 | 0 | 0.0 |
| Starlink Group 6-9 | 100 | 2023-119 | 22 | 1 | 4.5 |
| Starlink Group 6-10 | 101 | 2023-122 | 22 | 0 | 0.0 |
| Starlink Group 7-1 | 102 | 2023-124 | 21 | 0 | 0.0 |
| Starlink Group 6-11 | 103 | 2023-129 | 22 | 0 | 0.0 |
| Starlink Group 6-13 | 104 | 2023-131 | 22 | 0 | 0.0 |
| Starlink Group 6-12 | 105 | 2023-134 | 21 | 0 | 0.0 |
| Starlink Group 6-14 | 106 | 2023-138 | 22 | 0 | 0.0 |
| Starlink Group 7-2 | 107 | 2023-141 | 21 | 0 | 0.0 |
