# Peer review of "NORAD Tracking of the 2022 February Starlink Satellites and the Immediate Loss of 32 Satellites"

_Nonlinear Processes in Geophysics, 2024_

## Author Response (AR3)

September 9th, 2024.

Dear Editor,

Following the instructions sent by the NPG Editorial Office, we are submitting the point-by-point response to all the referee comments and the revised manuscript will all the changes marked in yellow.

The response to the referees are structured in the following way:

- comment from referees – text in black
- author's response – text in blue
- author's changes in manuscript – text in red

Once again, thank you for handling the revision of our manuscript and for your valuable comments.

Fernando Guarnieri

**RC1**: 'Comment on npg-2024-9', Alexandra Fogg, 07 May 2024

We would like to thank Dr. Fogg for carefully reading the manuscript and sharing helpful comments and suggestions. The manuscript will be revised accordingly. Please see below our response to each of your comments and suggestions.

*In this paper, the authors present Swarm density data during the Starlink loss event of February 2022. They also present some data showing the timeline of tracking by NORAD, including orbital altitudes. They conclude that none of the previously presented mechanisms can account for the satellite losses.*

*The paper is generally well-written with few typographical errors. I include major and minor comments below. Thank you for inviting me to review this paper, I would be happy to review it again if needed.*

- Thank you. We will carefully look for any typographical errors and correct them. All your major and minor comments will be addressed in the revised manuscript.
- Several typographical errors were corrected along the whole manuscript, as well as some missing explanation to abbreviations and some units/notations.

**Major comments**

*1 – Overall I felt the paper was a bit light on citations, and detailed description. I think in general a more detailed discussion of the implications/results of the work is needed to really emphasize the impact. I indicate some places where discussion needs expansion in the minor comments.*

- Thank you for pointing these out. We will improve the citations and descriptions in the manuscript to emphasize the impacts of the results.
- Some new citations and explanations were added. The precise location is indicated in the answer of each individual minor comment.

*2 - Just a note for the editor to consider. I am not sure whether this paper fits within the remit of NPG? On the NPG website (https://www.nonlinear-processes-in-*

*geophysics.net/about/aims_and_scope.html) is says the journal "solicits disruptive and innovative concepts and methodologies, as well as original applications of these to address the ubiquitous complexity in geoscience systems". Although this paper has merit in terms of new data etc, I am not sure if it is presenting any new concepts/methodologies. Perhaps the authors could comment on this.*

- Although this comment was directed to the Editor, we would like to emphasize that our article is pointing to an unexplained cause of a large loss of satellites. The physics behind the losses is not explained by any of the already proposed mechanisms. The real cause for such a large loss may lead to a new physical process acting in the high atmosphere that could be vital to our technologic space based devices. Such an event with so many spacecrafts would be a nice opportunity to understand the satellite loss mechanisms, since the individual data for all those spacecraft may provide a space resolution not seen before during a disturbed event.

- The text was changed mainly in Section 7 to make it clear.

**Minor comments**

*1 - Lines 19-26. Please could you consider including some more recent citations for storm work. For example, work by Walach et al (2019, https://doi.org/10.1029/2019JA026816) suggests an expansion to perhaps even 40deg latitude.*

- Thank you for the reference. This will be discussed in the revised manuscript.

- The suggested reference was added (line 419) as well a possible implication was discussed around line 280.

*2 - On line 34 you note that there are three published scenarios (Tsurutani, Dang, and Fang), but you don't describe them. Please briefly describe what each of those papers propose, since this is key to your paper.*

- The description of the mechanisms proposed by those papers was in the section 8, around line 240. But since you mentioned, we noticed that it may lack for a first-time reader of our article. We have included a short description of each one in the introduction. Thank you.

- We have included a description of the cited mechanisms in the beginning of the text, from lines 34 to 55.

*3 - Section 2 is quite light on references. You assert lots of facts about the scenarios – please could you include citations for these. If no papers are available, a webpage citation for where you got the information will suffice.*

- Thank you for the suggestion. We have included suitable citations to the stated facts.
- We included the references in the text (lines 62, 72, 74) and also in the citation list (lines 347-351).

*4 - Line 75: how did you determine the onset time of this flare / CME eruption? It appears to be from a list – please cite where you got it from.*

- The CME eruption is estimated based on the ICME velocity observed. This information is used to identify the flares observed by GOES X-ray sensor. We now have made it clear to the reader in the text and include the references.
- Included the information in lines 97-100 and the reference in line 380.

*5 - Line 79: did you investigate the driver of the second storm? Is it driven by the same CME event?*

- The interplanetary driver of the second storm is another magnetic cloud. A detailed discussion of the driver has been included. However, it was not possible to clearly identify a flare in the Sun that could be the origin of this event.
- The text was change between the lines 95 and 126 to make it clear.

*6 – Line 82: for sudden impulse, please briefly describe what this event is rather than assert it. You should also cite Araki 1994 (doi 10.1029/GM081p0183).*

- Thank you. The paper is now cited, and we included a short explanation about the SI in the text.
- The text was changed around line 105, included Araki (1994), and the reference was added in lines 337-338.

*7 - Lines 80-99. In your description of the passing solar wind / storm events, it's very factual. I think this description would not be clear to someone who isn't an expert in storms etc. If possible, please could you link your descriptions to the figure – e.g. on line 87 you note the fluxrope signature. Please describe what a characteristic fluxrope signature looks like (including citation) and link back to where you see this in your figure. Please do similar for each of the signatures you discuss.*

- Thank you for the suggestion. We have included a detailed definition and descriptions (with suitable references) for all major interplanetary structures.
- We included some brief description of the structures in lines 103-107, 112-113. Reference added in lines 380 and 388.

*8 - Fig 1: it could be worth including a vertical line / shaded region which indicates the Starlink launch window. Also, you could indicate storm phases from a published list (e.g. Walach 2019 mentioned above).*

- We have updated the figure accordingly, including an arrow on the top of the figure to indicate the time of the Starlink launch.
- An arrow was included on the top of the figure to indicate the Starlink launch time. The storm phases are shown directly from the SYM-H plot, where the decreasing values indicate the storm main phase, and subsequent increasing values indicate the recovery phase. We included marks in the figure to indicate the main and recovery phases.

*9 - Line 100. Please provide citations for the Swarm mission, data, and orbital characteristics.*

- The original reference for Swarm mission is "Swarm - The Earth's magnetic field and environment explorers. ESA report for mission selection (SP1269/6), April 2004". We have updated the text and included the reference in the paper.
- Citation include in line 128 and reference included in line 392, and indicated the website where the data is available.

*10 - Fig 2. Your choice of colour bar is not colourblind-friendly, and doesn't seem to be perceptively uniform. I would recommend changing the colourbar, and if not, at a minimum you could provide another panel which shows a timeseries of average density across some latitude window.*

- Thank you very much for your suggestion. We have processed the figures again to use a colormap colourblind-friendly.
- The Figures 1 and 2 were reprocessed using a colourblind-friendly colormap.

*11 - Line 125. I make it over 130% - you may as well state the exact percentage here.*

- We have included exact values of the dayside and nightside ionospheric densities, as suggested.
- Fixed on line 153.

*12 - Table 1: what is the cause of the 28.6% failure for launch number 76? Just a sentence with citation here as it's an obvious outlier with second biggest failure rate.*

- This launch (Group 6-1) had several changes compared with the previous satellites. It was the first launch of larger, upgraded Starlink V2 Mini satellites with four times the bandwidth of previous models (so the reduced number of satellite compared with the previous launches, around 50). Also, it was the first use of a Argon-fueled Hall-effect thruster in space. Space-X also made changes in the tension rods to avoid release them in space. So, with those large changes, the number of losses was larger. We have placed a mark in this launch and the explanation in the Figure Caption.
- This table was moved to the Appendix section as suggested by the Referee #2. We included a mark and an explanation (with references) in the table caption. The figure 3 caption also has a reference to this event. The references are in lines 381 and 423.

*13 – Line 155 "Some of these remaining 6 were also lost after a few tracking". This sentence doesn't quite make sense, please rephrase.*

- The statement has been revised to "Some of these 6 surviving satellites were also lost after a few tracking". Thank you.
- We rewrote the whole paragraph starting in line 184 to make it clear.

*14 - Fig 3: I think the caption should be more descriptive, including: noting the numbers are "NORAD" numbers, explaining what the different coloured arrows mean, noting that the timeseries is SYM-H, etc. Please follow the descriptive style of your previous figure captions.*

- Thank you for the suggestion. The figure caption has been updated/revised accordingly.
- The former Fig. 3 is now Fig. 4. A new more descriptive caption was included for the figure.

*15 - Fig 4 and 5: Please make the fontsize bigger, and perhaps make the figure the same width as the text. Lots of detail to be seen! It could be informative to put some indication of storm times on these panels (or perhaps just overplot SYM-H?).*

- Thanks for the suggestions. The figures have been updated, as suggested.
- The former figures 4 and 5 (now figures 5 and 6) were changed to increase the font size, with some other improvements in the information organization. However, the inclusion of SYM-H in the figures was meaningless, since in these cases, the first tracking occurred more than a day after the launch (see former figure 3 – now figure 4). The real SYM-H conditions that could be related to the losses occurred before the period of these orbit plots.

*16 – Line 221: typographical error: survived->survive*

- Corrected
- Fixed in line 256.

*17 - Line 222-223: you comment here about the wide variety of fates for the satellites. This is an interesting point. From my understanding of your introduction to Starlink launch procedures, they are ejected in different directions from the launch vehicle. Could that mean they are e.g. impacting the enhanced atmosphere at a different angle / time / velocity etc. Is there any correlation between the spacecraft that were lost and there position in the launch vehicle?*

- We certainly considered this possibility. However, it is not possible to connect each satellite to the stack where it was attached. The telemetry data from the first moment after the release could indicate the direction, but this data is not public.
- This was made clear in the new text between lines 265-309.

*18 - Line 235 – Please could you expand on this point rejecting Tsurutani 2022's mechanism. If there was an enhanced density at 500 km, could there be enhanced densities below as well?*

- Yes, this is possible.  But we feel that the density increase would be small and would not lead to a rapid loss of satellites.
- This was discussed again around lines 265-272.

*19 - Line 241 – missing year on Dang citation.*

- Corrected.
- Included the year in line 273.

*20 - Line 250 – 254. How does your work contribute to / back up this theory? If not, what does your study suggest is the cause of the loss?*

- Kakoti et al. (2023) showed different effects in Ionosphere and Thermosphere due to storms and substorms in different regions and altitudes. They used both ground and satellite observations, mainly TEC data, which are integrated in the whole column. In their conclusion, they also mention some possible mechanisms that could possibly lead to these disturbances ("Low-latitude ionospheric electric field/EEJ variation on 4 February could be related to the DDE, PPE field, and magnetospheric convection related to the substorm."). We believe that both our and their studies would benefit from local data from the telemetry of the lost satellites to pinpoint what is the driver of those changes in ionosphere, allowing us to create a complete scenario from the interplanetary disturbances to the ionospheric changes that lead to the satellite losses.
- The section between lines 290-309 discuss the possibilities of future studies to analyze this event with local data, not only modeling and simulation.

*21 - Section 8 – I would like to see a clear statement of what your work suggests the cause of the loss is. Could it be a combination of all the effects stated by previous authors? It was not an enormous storm, but did a bunch of small effects work together to create a tricky situation?*

- To answer this is beyond the scope of the present work. It is possible but in our current thinking it is not probable.  The very rapid loss of the Starlink satellites indicates that perhaps something more drastic has happened.

*22 - Line 262: the possibility of collisions causing the losses is remote – why? Citation for this? Could the geomagnetic conditions increase the possibility of this?*

- As mentioned before, it is a possibility but we cannot affirm this due to the lack of telemetry data. Private communication with people related to the business mention the probability exist but is small.
- In line 319 we mention that future studies may investigate this possibility. Only telemetry data would allow such this simple analysis.

**RC2**: ['Comment on npg-2024-9'](), Anonymous Referee #2, 10 Jul 2024

*The authors consider interplanetary and geomagnetic conditions during the launch of Starlink satellites as a potential cause of their loss in February 2022. The study is more a qualitative exercise and emphasising the need of deeper physical understanding of geomagnetic storm events. Although, the paper is well-written, I have concerns regarding novelty and used (qualitative) method. The explanation of the mechanisms that lead to the loss of the satellites remains vague. I was expecting more details and, in particular, a clear physical mechanism, described as a model and with equations. Without such clear description of the physical mechanism, the paper in its current state does not fit into the scopes of the journal NPG. For the moment, after reading I had the impression of a superficial summary from which I could not learn anything significantly new.*

- Thank you Referee #2 for reading our manuscript, and for the valuable comments. In our manuscript, we do mention that a complete model or an explanation of the mechanisms that lead to the losses would require data from the spacecrafts. Unfortunately, this data was not made public by the SpaceX. However, the already proposed mechanisms does not explain the losses, as we pointed out. The loss of such a large number of satellites, all them with their own telemetry, would be a valuable opportunity to analyze such effects from different positions in space. These different positions would allow us to understand whether the increased drag was locally generated or it was some structure propagating other regions. Even a limited set of telemetry data from each spacecraft, if made available by SpaceX, would be valuable for the scientific community, and the knowledge from these studies would be helpful for future SpaceX launches.

- We have made changes in the text, mainly around line 290-309, to make it clear.

*Some minor comments*

====================

*- the comparison with the other studies was mentioned in the beginning, but explained late in Sect. 8; it would be more helpful to have an understanding of these alternative explanations already from the beginning*

- This was also pointed by the other referee and we agree with the comment. We already made changes in the text to made it clear in the beginning of the text.

- Text changed from line 34 to 59

*- what about ESD events (Gubby and Evans, 2002; Wrenn, 1995; Iucci et al., 2005)?*

- Electrostatic Discharges (ESD) were not considered since SpaceX mentioned the satellites were functional and communicating until the reentering in the atmosphere. The increased drag experienced by the satellites was not possible to be compensated by the thrusters, leading to the loss of orbital stability.

- This discussion was included in lines 259-261.

  *- axes labels are too small*

- Thank you. We have already processed again the figures to make the labels bigger.

- Figures 5 and 6 were processed again to increase the labels sizes.

  *- colour schema in Fig. 2 is not colour-blind friendly*

- Thank you. The same issue was pointed by Referee #1, and we have already processed the figure again to change to colors to a more colour-blind friendly set.

- The Figures 1 and 2 were reprocessed using a colourblind-friendly colormap.

  *- Tab. 1 should be moved to appendix, but its main message should be included as a summarising figure (e.g., as a histogram)*

- Thank you for the suggestion. We will change it accordingly.

- Table moved to the Appendix, and we included a histogram as a Figure 3.

  *- several abbreviations are not explained (eg AU)*

- Thank you. We revised the text to explain all the abbreviations.

- This is corrected in line 102 as well as others around the text, such as in line 108.

  *- notation of the units should be checked (in particular of the superscripts)*

- Ok. Revised. Thank you.

- Several changes were made in the units and notations, mainly between lines 102 and 165.

**EC1**: 'Reply on AC2', Norbert Marwan, 17 Aug 2024

Thanks for the detailed response. Regarding my very first (and major) comment: if it is not possible to provide a model, I think it should be possible to provide some hypotheses or some ideas for future directions. I support the comment by Luis Vieira to "outline specific experimental or observational approaches that could be undertaken to identify and understand these mechanisms".

- Thank you for the question Dr. Marwan. Most of the already published and under review articles about these events are based in modeling and simulations. This time, we have the opportunity to have in situ observations from 49 satellites that experienced different conditions and fates. Some of them faced strong effects and were "killed" in the first hours or days after launch. Some survived a few more days and then failed, and the smaller number that survived after several days. We understand that private companies may be afraid of share their telemetry data in order to preserve their technology and business. However, if Starlink make public a minimum dataset of telemetry data, that would allow a multi-point data series analysis useful to understand the physics behind this event. This minimum dataset could have the position for each satellite, their velocities, altitudes, accelerometer data, and some indicative of whether the propulsion was activate, and the solar panels position.

  Knowing the position of the satellites and their velocity profiles (besides the accelerometer data) would indicate the region (in local time) where the most intense drag increases occurred. That could be in the midnight sector, due to some effect of the magnetosphere tail reconnection, although not expected in such low latitudes for a moderated storm. The increases in the dayside could indicate some change in the ionosphere induced by electric field penetration (affecting not only the ionosphere, but with possible effects on neutrals as well). Tsurutani et al. (2004) had already demonstrated the PPEF occurrence in higher altitudes (around 500 km). However, the plasma drag to higher altitudes may lead to the repopulation of other ions or even neutrals in lower altitudes in a dynamic not completely understood.

  The velocity profiles along these inclined orbits could show us, by the comparison among the satellites, whether there is a propagation of effects from high latitudes in the polar region to lower/mid latitudes. Those data would also help to confirm the observations by Walach and Grocott (2019), that during the geomagnetic storms, the auroral fluxes may expand equatorward as low as 40°, lower than previously expected as limit to 50° and higher. Also, these observations would help us to understand whether these regions with increased drag were continuous regions or are "tongues" of increased densities, and whether this phenomena could spread in all local times or are restrict to specific regions.

  Even the possibilities of collisions between satellites during the first moments after launch could be confirmed or discarded by a quick analysis of their accelerometer data.

  In despite the sad event for SpaceX loosing such a large number of satellites, a simple dataset from these units would help the whole community of ionosphere and geophysics. It would provide rare multi-point local data, during a disturbed period, and in a region where

the satellites usually stay for a very short period of time before being propelled to their final and higher orbits.

- These considerations were included in the manuscript between lines 290-309.

**CC1**: ['Comment on npg-2024-9'](), Luis Vieira, 16 Jul 2024

*Comparison with Previous Events: The paper briefly compares the February 2022 Starlink satellite losses with previous events. Can the authors elaborate on how the geomagnetic conditions and satellite responses during this event differ from or are similar to past satellite losses during geomagnetic storms? Are there any specific lessons learned from past events that could be applied to this case?*

- Previous space weather events related to satellite losses occurring during intense geomagnetic storms have been widely analyzed for tens of years. The main difference in this event is the occurrence of such a wide loss during an only moderated storm has never been seen before. Of course, the decision of SpaceX to initially inject the satellites in a low orbit was most likely related to the loss process. If we had more Starlink satellite loss information (from different launches) that would be a big help.

  The main lesson learned from this episode is that even a moderated storm may lead to losses when dealing to such a low altitude (in the limit of orbital conditions). In these cases, it would be essential to really know all the effects that could be affecting the satellites. The lack of an explanation during this event, may lead to new losses in the future, mainly due to the large number of satellites and constellations being launched.

*Future Research Directions: The paper calls for further investigation into the unknown physical mechanisms causing significant drag on satellites. Can the authors outline specific experimental or observational approaches that could be undertaken to identify and understand these mechanisms? What role might new satellite missions or advancements in space weather modeling play in this research?*

- For the satellites operating in orbits from 300 km and above, the vast number of satellites with public data allowed that researchers, along decades, elucidate the mechanisms acting during space weather events, with local sources of increased densities, or with increases in densities in high altitude that may propagate to lower latitudes.

  However, the launches in such a low orbits, below 250 km, still lack the data to perform analysis and studies. The scientific community would be greatly benefitted if companies operating in such low orbits with a large number of satellites could make public telemetry datasets (although protecting their sensitive information). The large number of satellites almost evenly distributed would allow the mapping of the effects in their spatial distribution and also its evolution (drift) in time.

- The answers to these comments were included in the manuscript between lines 290-309.

*CC2*: 'Comment on npg-2024-9', Luis Vieira, 09 Aug 2024

*Revise the number of satellites in each stack. In this launch, there were 24 or 25 satellites in each stack.*

- Thank you for your comment. The correct number is really 24 or 25 satellites in each stack for this launch. We fixed the text already.

- Text fixed in line 66.

Reply to Report #1

Comment:

The authors have revised their manuscript. I am still concerned on the suitability of the paper for the journal. The journal aims and scope state that it "solicits disruptive and innovative concepts and methodologies, as well as original applications of these to address the ubiquitous complexity in geoscience systems...". At present, I don't see neither a clear application of such methodologies nor the inclusion of innovative concepts. Therefore, please consider to incorporate or at least discuss some advanced analysis or concepts that emphasize nonlinearity, making it more aligned with NPGs standards. This addition wouldn't necessarily require a lot of work.

*Dear Dr. Norbert Marwan,*

*Your last question in the interactive discussion was posted just after we had uploaded our revised manuscript. We are happy to have the opportunity to include our comments in this version. We used this opportunity to correct also some typos and check cleanliness of some sentences. Bellow is our answer to your previous comments:*

*Although our paper makes it clear that the standard satellite drag computer codes are not able to predict the fast loss of 32 of the Starlink satellites, perhaps including the possible nonlinearity of the atmospheric changes during disturbed geomagnetic periods, and the satellite drag with spacecraft orientation would help. Present models assume satellites with standard cross sections. However, the Starlink satellites are formed by two flat panels (one for the satellite body and another for the solar panels). Just after the launch, the solar panels are folded on top of the satellite panel. During the period for the orbital elevation, the solar panels are unfolded but kept aligned with the satellite body, becoming a long flat panel. In the final position, the satellite assumes an "L" shape. This unusual shape of Starlink satellites, make their cross sections completely different than traditional satellites, and highly dependent of its orientation.*

*Future computer codes should take these different shapes into account, their orientation, and then calculate a maximum and minimum time for reentry. The codes should also consider satellite tumbling. Would the loss times be within the max and min entry times or would they be even greater?*

*We include this discussion in the revised version of the manuscript, around lines 318 and 328.*

Reply to Report #2

Review for "NORAD Tracking of the 2022 February Starlink Satellites and the Immediate Loss of 32 Satellites" by Guarnieri et al, submitted to NPG

The authors have made some changes, although not all of my original points have been addressed. I still feel that the paper is overall light on citations and detailed explanation. I detail this below. Throughout this report, I use the point numbers from my original report.
Thank you for inviting me to review this paper.

*Dear Dr. Alexandra Fogg,*

*We thank you for your constructive comments. We have made additional changes to clarify the points you considered still missing. We also took this opportunity to correct some typos and improve sentence clarity. Below are our responses to your comments.*

Major Comments

2. I still feel that this paper is not within the remit of NPG, and note that Anonymous Referee #2 also commented on this. This is up to the editor to decide.

*As we mentioned in the answer to Referee #2, the last round in the interactive discussion occurred after we had uploaded the revised version of the manuscript. Now we had the opportunity to include this discussion of future steps around lines 318 to 328.*

Minor Comments

1. Although my suggested citation was included in the Discussion section, no more citations were added to the introduction on recent storm work. My view is that the introduction should include both original/seminal citations, as well as more recent, novel work.

*We included new citations:*

*line 110: Fiori et al., 2014; Lühr et al., 2009; Oliveira & Samsonov, 2018; Takeuchi et al., 2002;*

*lines 113, 114 e 115: Burlaga et al. 1998; Gopalswamy et. al 1998; Lepri and Zurbuchen, 2010; Sharma and Srivastava, 2012; Gopalswamy, 2015, 2022*

*line 119: Yashiro et al., 2013; Marubashi et al., 2015;*

*line 123: Wang et al., 2018; Kozyra et al., 2013.*

*These references were included and highlighted in the references list.*

3. The authors have added two citations (Clark+ 2022a and 2022b) to this section. My feeling is there are still many asserted facts without citations. For example lines 65-69 have no citations. If all this information is from Clark+, that's probably ok, but please make this clear.

*We adjusted the citations in line 65 (Manley, 2021) and in line 68 (Clark, 2022b and McDowell, 2023);*

7. I am happy with the flux rope explanation, but my view is that not all the phenomena have been described clearly enough. For example line 114-115 "Thus, the magnetic storm is caused...". This needs expanding – a negative peak in SYM-H isn't all that's needed for a geomagnetic storm, it needs to be a characteristic shape as well.

*We included the following explanation in between the lines 77 and 82:*

*"For the geomagnetic storm identification, we followed the classical definition by Gonzalez et al., (1994), which uses the Dst index (equivalent to the SYM-H index used here) to identify a storm. When the index falls in the range from -30 nT to -50 nT for more than 1 hour, it characterizes an small (weak) storm. Events with decreases between -50 nT and -100 nT for more than 2 hours are considered a moderate storm. An intense storm present falls to Dst values of less than -100 nT for more than 3 hours. The recovery phase is the period when the index returns to the values observed prior the storm. It may last from hours to several days."*

17. I like your explanation in the response to reviewers document, however I can't find the similar explanation in the new text. It's worth just taking your sentences from response to reviewer and putting them into the draft.

*We checked again our previous answers to include in the manuscript.*

*Line 100 to 105;*

*Line 310 to 312;*

*Lines 318 to 328.*

18. You say in the text on line 273 "We think this is not probable". Please justify this in the text.

*In the original publications of PPEF, it was shown these effects occurring uplifting plasma from about 300 km to altitudes as high as 600 km. These altitudes were demonstrated by satellite data (CHAMP, SAC-C and TOPEX). Since the SpaceX satellites never reached such high altitudes, we believe they were not directly affected by PPEF. Anyway, we recommended future simulations to investigate whether a similar effect could be happening in lower altitudes.*

*The text was changed in between lines 275 and 280.*

21. I don't feel the authors have addressed my point here. I would like to see a clear conclusion point, saying what steps forward their analyses have caused. I think it is necessary and within the scope of the paper to say what results they have found in terms of which mechanism their work contributes to.

*A new discussion was included since the last revision related to the other referee discussion. It is between the lines 318 e 328.*

22. I understand you cannot get the data from the company – this is of course a shame in terms of our community's drive towards space weather preparedness. Perhaps you could add a statement in section 8 to the effect of emphasizing the need for this telemetry data in our community.

*Sure. The statement was included between the lines 341 to 349.*